# Correlating qRT-PCR, dPCR and Viral Titration for the Identification and Quantification of SARS-CoV-2: A New Approach for Infection Management

**DOI:** 10.3390/v13061022

**Published:** 2021-05-28

**Authors:** Martina Brandolini, Francesca Taddei, Maria Michela Marino, Laura Grumiro, Agata Scalcione, Maria Elena Turba, Fabio Gentilini, Michela Fantini, Silvia Zannoli, Giorgio Dirani, Vittorio Sambri

**Affiliations:** 1Unit of Microbiology, The Great Romagna Hub Laboratory, 47522 Pievesestina, Italy; martina.brandolini@outlook.it (M.B.); fra.taddei@hotmail.it (F.T.); mariamichela.marino@auslromagna.it (M.M.M.); laura.grumiro@auslromagna.it (L.G.); agata.scalcione@auslromagna.it (A.S.); michela.fantini@auslromagna.it (M.F.); silvia.zannoli@auslromagna.it (S.Z.); giorgio.dirani@auslromagna.it (G.D.); 2Xenturion Srl, 47122 Forlì, Italy; info@xenturion.it; 3Department of Veterinary Medical Sciences, Alma Mater Studiorum—University of Bologna, Ozzano dell’Emilia, 40064 Bologna, Italy; fabio.gentilini@unibo.it; 4Department of Experimental, Diagnostic and Specialty Medicine—DIMES, Alma Mater Studiorum—University of Bologna, 40138 Bologna, Italy

**Keywords:** SARS-CoV-2, COVID-19, qRT-PCR, Ct, dPCR, TCID50/mL, viral titration, RNA copies

## Abstract

Severe acute respiratory syndrome coronavirus 2 (SARS-CoV-2) was first identified in Wuhan, China, in late 2019 and is the causative agent of the coronavirus disease 2019 (COVID-19) pandemic. Quantitative reverse-transcription polymerase chain reaction (qRT-PCR) represents the gold standard for diagnostic assays even if it cannot precisely quantify viral RNA copies. Thus, we decided to compare qRT-PCR with digital polymerase chain reaction (dPCR), which is able to give an accurate number of RNA copies that can be found in a specimen. However, the aforementioned methods are not capable to discriminate if the detected RNA is infectious or not. For this purpose, it is necessary to perform an endpoint titration on cell cultures, which is largely used in the research field and provides a tissue culture infecting dose per mL (TCID50/mL) value. Both research and diagnostics call for a model that allows the comparison between the results obtained employing different analytical methods. The aim of this study is to define a comparison among two qRT-PCR protocols (one with preliminary RNA extraction and purification and an extraction-free qRT-PCR), a dPCR and a titration on cell cultures. The resulting correlations yield a faithful estimation of the total number of RNA copies and of the infectious viral burden from a Ct value obtained with diagnostic routine tests. All these estimations take into consideration methodological errors linked to the qRT-PCR, dPCR and titration assays.

## 1. Introduction

In late December 2019, a series of atypical pneumonia cases of probable infectious origin were reported in Wuhan (Hubei Province, China). Soon after the isolation of the pathogen, Chinese authorities confirmed that the febrile respiratory disease was of viral etiology and identified as the causative agent a novel coronavirus, renamed severe acute respiratory syndrome coronavirus 2 (SARS-CoV-2) due to the high sequence homology with another zoonotic β-coronavirus, SARS-CoV. The disease caused by SARS-CoV-2 is known as coronavirus disease 2019 (COVID-19) [1,2,3,4].

To date, SARS-CoV-2 is the third zoonotic coronavirus that was able to cross the species barrier and infect humans causing severe respiratory infections in less than two decades, along with severe acute respiratory syndrome coronavirus (SARS-CoV) and Middle East respiratory syndrome coronavirus (MERS-CoV) [5,6]. The severe acute respiratory syndrome coronavirus which was first identified in the Guangdong Province, China, in late 2002, was responsible of the first coronavirus epidemic of the 21st century, infecting approximately 8000 people and leading to at least 774 deaths (10% death rate) [7,8,9]. In 2012, the Middle East respiratory syndrome coronavirus emerged in Jordan; since then, over 2500 cases have been confirmed and nearly 860 patients died due to the infection and its complications (35% death rate) [10,11]. Both epidemics were successfully contained as the transmission chain was swiftly interrupted.

Unlike other human coronaviruses, it has been widely documented that SARS-CoV, MERS-CoV and SARS-CoV-2 have a much higher pathogenicity and lethality due to their tendency to infect the lower respiratory tract, resulting in lung injury and acute respiratory distress syndrome (ARDS) which add up to septic shock and can rapidly lead to multiple organ failure and consequently death [12,13].

Despite some common characteristics shared by these three viruses with particular regard to the animal reservoir (all of them likely derive from bat coronaviruses [2,14,15,16,17,18]), the mechanisms underlying their pathogenicity, the route of human-to-human transmission and, at least to a certain extent, the overlapping range of clinical manifestation [19,20], the novel coronavirus possesses some unique features (notably, very high transmissibility) that enabled its global uncontrolled spread regardless of the containment measures. On 11 March 2020, the World Health Organization declared COVID-19 a pandemic; as of today, SARS-CoV-2 accounts for more than 100 million confirmed cases and about 2 million deaths in 219 countries [21].

Quantification of the viral load in a virus-containing solution, whether it be a clinical sample or a cell culture supernatant aliquot, is an obligatory passage in routine diagnostics as well as in research.

A traditional method to quantify the infectious titer is by endpoint titration and calculation of the median tissue culture infecting dose per mL (TCID50/mL), defined as the burden of viral infectious particles per unit volume capable of producing a cytopathic effect in half of the infected cell cultures [22,23,24]. This process demands an extensive knowledge in cell culture propagation and maintenance, not to mention that it is laborious and requires three to four days on average to complete. Furthermore, cell infection must be carried out in a Biosafety Level 3 laboratory (BSL-3). Considering all the above reasons, in certain circumstances, an endpoint titration may not be viable as a consequence of the lack of one of the minimum aforementioned requirements or, simply, may not be a convenient option. Another aspect to consider is the variability connected with the use of a biological system, which could easily affect the results, making this kind of quantification difficult to standardize. In fact, the biological variation of the system is high due to the difficulty in plating the same number of cells, adding the same amount of virus, stopping the infection at the same time. In all these cases, nucleic acid detection and amplification strategies, namely a semi-quantitative and quantitative reverse-transcription polymerase chain reaction using a TaqMan fluorescent probe, may yield to an equally sensitive quantification of the viral burden in a specimen of interest without requiring biocontainment measures and shortening the time of analysis.

Semi-quantitative and quantitative methods differ for the precision of the result they return at the end of the analysis. In semi-quantitative methods (quantitative reverse-transcription polymerase chain reaction (qRT-PCR)), the intensity of fluorescence emitted is recorded after every amplification cycle to determine the relative quantity of target and the cycle threshold values (Ct) thus obtained are inversely proportional to the magnitude of the viral load, but they cannot be directly converted into genome copy number equivalents. A relative quantification of target copies employing qRT-PCR presupposes the generation of a standard curve by amplification of calibrators containing a known amount of target. By contrast, in quantitative methods, specifically digital PCR (dPCR), the sample is partitioned in thousands of independent reactions so that we can probabilistically assume that target molecules are randomly distributed and each individual reaction mixture contains at most one target; the number of partitions that emit fluorescence at the end of the amplification directly equals the number of target copies present in the sample, thereby providing an absolute quantification without depending on calibration to interpret the results. This aspect is of particular concern when standard samples are not available [25,26].

Establishing whether there is a correlation between the results of such different quantification methods would allow for time saving as well as efficient determination of the total number of genome copies present in a given specimen and, more important, an accurate calculation of the viral titer starting from Ct values, easily obtained from a qRT-PCR, hence normalizing Ct results to an objective and exact determination of the actual number of infectious virions.

## 2. Materials and Methods

### 2.1. Cell Line and Virus 

Quantification of the viral load by endpoint dilution was accomplished using Vero E6 cells (ATCC CRL-1586), a continuous line isolated from African green monkey (formerly known as *Cercopithecus aethiops*) kidney epithelium as it is widely documented that they are sensible and permissive to SARS-CoV-2 infection, leading to high titer replication [27,28,29]. Cells were maintained in Dulbecco’s modified Eagle’s medium (DMEM) supplemented with 10% heat-inactivated fetal bovine serum (FBS), 1% penicillin-streptomycin (P/S) and 1% L-glutamine (L-Gln). Culture medium and supplements were all purchased from EuroClone (Milan, Italy).

The correlation between qRT-PCR and dPCR was obtained using residual clinical specimens, (human nasopharyngeal and oral swabs) submitted to the Unit of Microbiology, Greater Romagna Area Hub Laboratory, Cesena, Italy, for diagnostic purposes. On the other hand, the correlation between molecular methods and endpoint titration was achieved using a viral strain isolated from a single residual sample (nasopharyngeal swab) taken from a patient with laboratory-confirmed COVID-19 and propagated in Vero E6 cells. The obtained viral stock has been aliquoted and stored at −80 °C. The sample was analyzed employing the FilmArray Respiratory Panel (Biomerieux, Marcy l’Etoile, France) to exclude the presence of other viruses which can cause cytopathic effect of Vero E6 cells. After the propagation, the viral stock was re-tested using the same assay, resulting negative. Furthermore, the viral strain was sequenced using CleanPlex SARS-CoV-2 Flex (Paragon Genomics, Inc., Hayward, CA, USA) and Illumina MiSeq (Illumina Inc., San Diego, CA, USA). The software analysis is Sophia Genetics (Lausanne, Switzerland) which identified a B.1 lineage of SARS-CoV-2 (GISAID code EPI_ISL_1908157).

### 2.2. Viral Titration

The day prior to infection, approximately 20,000 cells per well were seeded in 96-well tissue culture plates (about 2,000,000 cells per plate) using 5% FBS DMEM, and then incubated overnight at 37 °C in a humidified, 5% CO_2_ atmosphere-enriched chamber.

On the day of infection, serial 10-fold dilutions (from 10^−1^ to 10^−7^) of the viral stock (isolated and propagated as discussed above) were prepared in 2% FBS DMEM and used to infect a confluent monolayer of cells; each dilution was tested in eight replicates. Dilutions ranging from 10^−1^ to 10^−5^ were subsequently titrated using the same approach. In every plate, four wells were used as no-virus control and four more wells were used as virus control. Undiluted viral stock and dilutions were titrated in duplicate. The plates were incubated for 72 h and observed daily to monitor the development of cytopathic effect (CPE) employing an inverted optical microscope. On day three post-infection cell culture supernatant was removed and cells were fixed and stained by means of a 4% formaldehyde (Fisher Chemical, Milan, Italy) solution in crystal violet (Delcon, Bergamo, Italy) incubated for 30 min at room temperature. Cytopathic effect was evaluated and recorded.

Viral titers, expressed as TCID50/mL, were calculated according to both Reed and Muench and Karber methods based on eight replicates for dilution, as previously described [30,31,32]. 10^−6^ and 10^−7^ dilutions could not be successfully titrated because of a lack of data.

### 2.3. Nucleic Acid Quantification 

Aliquots of each viral dilution were processed using two different semi-quantitative nucleic acid amplification method. Each dilution and undiluted viral stock were tested in six replicates with every system.

#### 2.3.1. Quantitative Reverse-Transcription Polymerase Chain Reaction (qRT-PCR)

Allplex SARS-CoV-2 Extraction-Free (Seegene Inc., Seoul, Korea) is a real-time qRT-PCR assay which does not require a preparatory RNA-extraction, but rather relies on the thermal lysis taking place during the reverse transcription reaction in which the specimen is warmed up to 50 °C for 20 min (reverse transcription) and then to 95 °C for 15 min (polymerase activation). The assay enables the simultaneous detection of three target genes, namely the E gene (common to all Sarbecoviruses), RdRP/S gene and N gene (specific for SARS-CoV-2). Sample preparation, reaction setup and analysis were performed according to the manufacturer’s instructions. In brief, 15 µL of each sample were diluted 1:4 in 45 µL of RNase-free water in a 96-well PCR plate and hence 5 µL were transferred to another plate with 16 µL of PCR master mix, containing 5 µL of MOM (MuDT Oligo Mixture, containing dNTPs, oligos, primers and TaqMan 5’ fluorophore / 3’ Black Hole Quencher probes), 5 µL of enzymes, 5 µL of RNase-free water and 1 μL of internal control for every reaction. A positive and a negative control were included. The assay was run on a CFX96 real-time thermal cycler (Bio-Rad, Feldkirchen, Germany). The amplification process includes a first step for cDNA denaturation at 95 °C for 10 s, followed by primers annealing at 60 °C for 15 s and elongation at 72 °C for 10 s (44 cycles). Fluorescent signals were acquired after every amplification cycle for FAM (E gene), Cal Red 610 (RdRP/S gene), Quasar 670 (N gene) and HEX (internal control) fluorophores. Results analysis and targets quantification were carried out employing the 2019-nCoV viewer from Seegene Inc.

Nextractor (Genolution Inc., Seoul, Korea) is an automated extraction system which allows a rapid and efficient purification of viral RNA prior to reverse transcription and amplification by employing silica magnetic beads in the presence of high concentrations of chaotropic salts. For the reaction, 200 µL of sample and 10 µL of internal control were added to the lysis buffer already present in the plate. Eluates were, thereafter, transferred in a 96-well PCR plate with 15 µL of PCR master mix, containing 5 µL of MOM (MuDT Oligo Mixture), 5 µL of enzymes and 5 µL of RNase-free water. A positive and a negative control were included in each plate. The analysis was performed following the Allplex SARS-CoV-2 protocol from Seegene previously described.

#### 2.3.2. Digital Polymerase Chain Reaction (dPCR)

dPCR was separately performed on 48 residual UTM (Universal Transport Medium, COPAN Italia S.p.A., Brescia, Italy) from positive clinical samples (nasopharyngeal and oral swabs). For diagnostic purposes, these samples had been processed using Nextractor for the RNA extraction and with Seegene Allplex SARS-CoV-2. Specimens with low (<20), intermediate (20–25) and high (>25) Ct values were included. All specimens were stored a maximum 24 h at 2–8 °C until processing. For the aim of the present work, a second viral RNA purification was carried out using the aforementioned Nextractor extraction system. The RNA extracted was divided in two aliquots. One aliquot of each eluate was analyzed following the Allplex SARS-CoV-2 protocol from Seegene. A second aliquot of eluate was preserved in dry ice for the transport (less than 20 min) to Xenturion Laboratory and directly reverse transcribed employing the iScript cDNA Synthesis kit (Bio-Rad). In brief, 10 µL of each extract were mixed with 4 µL of iScript Reaction mix (containing MMLV RNase H+ reverse transcriptase, dNTPs, oligo(dT)s and random primers), 1 µL of iScript Reverse Transcriptase and nuclease-free water to a final volume of 20 µL. The retro-transcription reaction, run on a thermal cycler, comprehends 5 min at 25 °C for priming, 20 min at 46 °C for reverse transcription and 1 min at 95 °C for reverse transcriptase inactivation. cDNA thereby obtained was consequently used as a template for dPCR. 1.5 µL of every cDNA template were added to 1× QuantStudio 3D Digital PCR Master Mix v2 (Thermo Fisher Scientific, Monza, Italy), 500 nM of forward and reverse primers, 125 nM of TaqMan 5’ FAM / 3’ Black Hole Quencher probe and nuclease-free water to reach a final volume of 15 µL. Primers and probes employed for the assay were designed on the conserved region N1 of nucleocapsid gene and purchased from from Integrated DNA Technologies as listed in the CDC (Centers for Disease Control and Prevention) protocol https://www.cdc.gov/coronavirus/2019-ncov/downloads/rt-pcr-panel-primer-probes.pdf (accessed on 1 December 2020). The reaction mix was subsequently transferred on a QuantStudio 3D Digital PCR 20K v2 Chip. The amplification reaction was run on a ProFlex 2 × flat PCR System thermal cycler following a protocol, which includes 10 min at 96 °C for initial cDNA denaturation, followed by 30 s at 60 °C for annealing and elongation and 2 min at 98 °C for denaturation (39 cycles), and a final step at 60 °C for 2 min for final elongation. Following thermal cycling, image capture and data analysis was performed using QuantStudio 3D Digital PCR Instrument and QuantStudio 3D AnalysisSuite Software (Thermo Fisher Scientific, Monza, Italy).

The analysis of 48 specimens using dPCR and a traditional qRT-PCR method has been used to obtain a statistical correlation between the two methods. Subsequently, the equation extrapolated from the graph has been utilized to calculate the number of RNA copies of the viral stock and its dilutions.

All assays involving potentially infectious SARS-CoV-2 were performed in a BSL-3 laboratory at the Unit of Microbiology, Greater Romagna Area Hub Laboratory, Cesena, Italy. Purified RNA samples for dPCR were sent to Xenturion laboratory in Forlì, Italy. 

All graphs were generated using GraphPad Prism 9 (GraphPad Software Inc., San Diego, CA, USA) [32].

## 3. Results

### 3.1. Comparison of qRT-PCR Methods

Ct values were obtained by processing each viral dilution with the two amplification methods (i.e., performing a preliminary RNA extraction or directly submitting the sample to amplification) were graphed against the dilution (on a log10 scale); Ct equivalents showed linearity across the considered concentration range (Nextractor: y = −3.293x + 10.91, SE (slope) = 0.03575, SE (Y-intercept) = 0.1289, r^2^ = 0.9953, *p* value < 0.0001; Allplex SARS-CoV-2 Extraction-Free: y = −3.324x + 15.79, SE (slope) = 0.06450, SE (Y-intercept) = 0.2326, r^2^ = 0.9852, *p* value < 0.0001) (Figure 1).

Considering the slope values of the lines we then compared the two sets of data. This analysis demonstrated that there is a correlation between the recorded results (y = 0.9907x–4.733, SE (slope) = 0.01103, SE (Y-intercept) = 0.2935, r^2^ = 0.9951, *p* value < 0.0001) (Figure 2). Ct values are presented as the mean of six replicates ± SE.

### 3.2. Calculation of Tissue Culture Infecting Dose per mL (TCID50/mL)

Titers of the viral stock and 10-fold serial dilutions from 10^−1^ to 10^−5^ were estimated by endpoint titration and calculation of the TCID50/mL according to both the Reed and Muench and Karber methods. Titration has been undertaken in duplicate in separate days by different operators. Resulting titers for both methods were then graphed as a function of the dilution (on a log10 scale), showing linearity across the considered concentration range (Reed and Muench: y = 1.035x + 6.546, SE (slope) = 0.03007, SE (Y-intercept) = 0.09104, r^2^ = 0.9916, *p* value < 0.0001; Karber: y = 1.042x + 6.567, SE (slope) = 0.03447, SE (Y-intercept) = 0.1044, r^2^ =0.9892, *p* value < 0.0001) (Figure 3). Furthermore, the differences among the mean values obtained with Reed and Muench and Karber methods have been analyzed with a Student’s *t* statistical test which has shown no statistically significant difference (*p* value = 0.91).

### 3.3. Correlation between dPCR and qRT-PCR

Copies/µL (on a log10 scale) estimated by dPCR were graphed as a function of qRT-PCR Ct values obtained processing samples with preliminary extraction with Nextractor and following amplification accordingly to the Allplex SARS-CoV-2 Extraction-Free protocol. Ct value of RT-PCR was highly correlated with the copy number determined by dPCR (y = −0.3062x + 10.50, SE (slope) = 0.01427, SE (Y-intercept) = 0.3596, r^2^ = 0.9127, *p* value < 0.0001) (Figure 4). However, as shown in the graph, for Ct values above 27 the correlation decreases considerably due to the presence of a limited quantity of RNA.

### 3.4. Correlation between qRT-PCR and TCID50/mL

qRT-PCR Ct values obtained processing samples with preliminary extraction with Nextractor and following amplification according to the Allplex SARS-CoV-2 Extraction-Free protocol were further correlated with TCID50/mL calculated according to the Reed and Muench method which is mostly used (y = −0.3152x + 9.988, SE (slope) = 0.009107, SE (Y-intercept) = 0.1816, r^2^ = 0.9917, *p* value < 0.0001) (Figure 5).

## 4. Discussion

The major difference between quantification of the viral load by endpoint titration, on the one side, and quantification by nucleic acid detection and amplification strategies, on the other, is their accuracy. In the first place, Ct values from semi-quantitative methods can only be considered as a macroscopic approximation of the real viral burden, they are system- and reaction-specific, and closely depend upon the threshold, either automatically or manually set (these characteristics must be taken into consideration also when deciding to plot a calibration curve for relative quantification). In fact, Ct values are dependent upon the extraction/purification method as well as the amplification efficiency of the PCR assay. One of the purposes of this paper was to evaluate if there is a correlation between a method with nucleic acid extraction/purification and a method without it, maintaining the same reagents and protocol of amplification. As emerged from the analysis of the results (Figure 1), there is a statistically significant relation between these two methods. Clearly, for the same sample, the amplification of extracted and purified RNA provides lower Ct values compared to the extraction-free method, because in the latter the reaction of amplification is affected by the presence of all the other cell components that decrease the overall sensibility of the PCR (higher Ct values).

Secondly, endpoint titration and molecular analysis differ for the ability in distinguishing infectious viral particles from non-infectious RNA, virtually present in every specimen. With the first method, we measure the ability of the virus allegedly present in the specimen to produce an appreciable cytopathic effect, which can only be attributable to infectious viral particles. The latter, on the contrary, could lead to an overestimation of the titer by detecting the total amount of RNA molecules, including both genomic RNA and subgenomic mRNAs, rather abundant in clinical samples as well as in cell cultures, as demonstrated in a previous study [33]. This observation is valid both for qRT-PCR and dPCR, which ensure high sensibility but comparatively lack specificity. Hence, to fill this gap, we decided to correlate the PCR results with TCID50/mL which is a measure of the RNA that belongs to viable virions able to infect cells and produce a visible cytopathic effect.

The workflow shown in Figure 6 summarizes how we decided to structure our project, what kind of samples were used, the methodologies employed, the results we obtained and the converter sheet we could finally draw from them.

Our data demonstrate a strong correlation among TCID50/mL, Ct values and number of RNA copies per µL, units of measure commonly used to provide a quantification of the viral load in diagnostic and research fields. These findings have been concretized in a converter sheet capable of returning an estimated value of RNA copies and TCID50/mL for a given Ct value obtained with the Allplex SARS-CoV-2 Extraction-Free protocol, routinely used in diagnostics. Every calculated output takes into consideration the statistical error related to each method. TCID50/mL thus determined can only be considered as a mere estimation due to the intrinsic variability of the method and the operator, as already discussed in the introduction of the present article.

In this context and against this background, molecular methods, with their sensibility, reproducibility, wider laboratory application, time- and cost-effectiveness can meet the need for a rapid and accurate determination of the infectious viral titer, hence representing a good compromise for every diagnostic or research laboratory.

Although the methods we described are well known and have been already applied either to management of SARS-CoV-2 infection or to research purposes, their integration constitutes a novel approach which has never been described. The strength of this combination is that it allows a more global view among alternative but different techniques.

In the future, it would be of considerable interest to investigate this correlation among other types of methods and materials, such as inactivated samples and swabs with different Ct values stored in distinct conditions that can affect the viability of the virus.

## Figures and Tables

**Figure 1 viruses-13-01022-f001:**
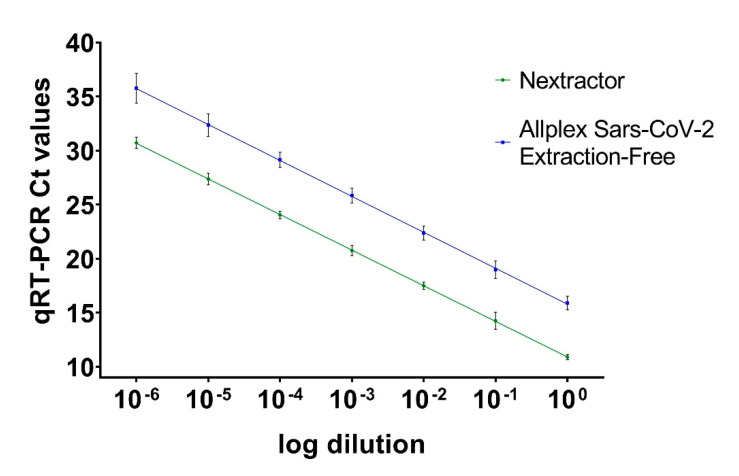
Ct values obtained with preliminary RNA extraction (Nextractor shown in green line) and the extraction-free protocol (Allplex severe acute respiratory syndrome coronavirus 2 (SARS-CoV-2) Extraction-Free, shown in blue line). Ct values are presented as the mean of six replicates for each dilution ± standard error (SE).

**Figure 2 viruses-13-01022-f002:**
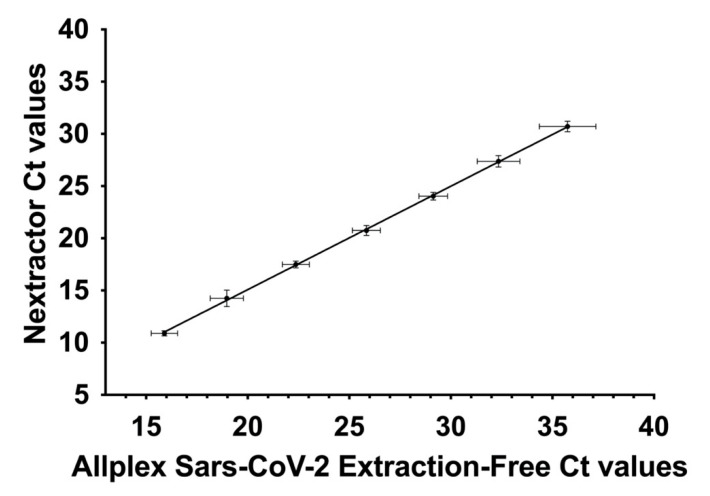
Correlation between Ct values obtained with preliminary RNA extraction (Nextractor) and the extraction-free protocol (Allplex SARS-CoV-2 Extraction-Free). Ct values are presented as the mean of six replicates for each dilution ± SE.

**Figure 3 viruses-13-01022-f003:**
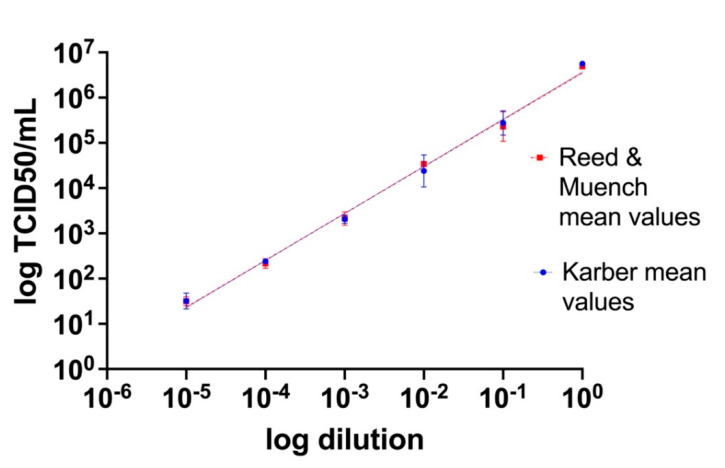
Tissue culture infecting dose per mL (TCID50/mL) values obtained by titration on Vero E6 cell cultures of scalar dilutions of the viral stock. TCID50/mL values are presented as the mean of the two different methods (Reed and Muench and Karber) of two replicates for each dilution ± SE.

**Figure 4 viruses-13-01022-f004:**
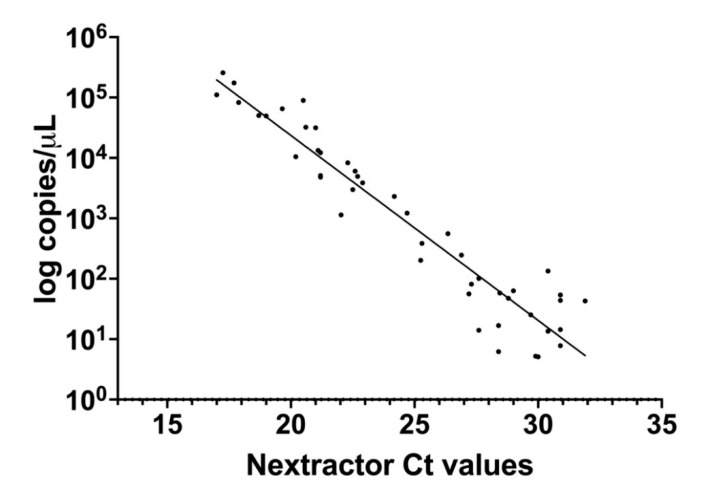
Correlation between the results obtained from the same set of samples by digital polymerase chain reaction (dPCR, number of RNA copies per µL on a log 10 scale) and quantitative reverse-transcription polymerase chain reaction (qRT-PCR) with preliminary RNA extraction (Nextractor Ct values).

**Figure 5 viruses-13-01022-f005:**
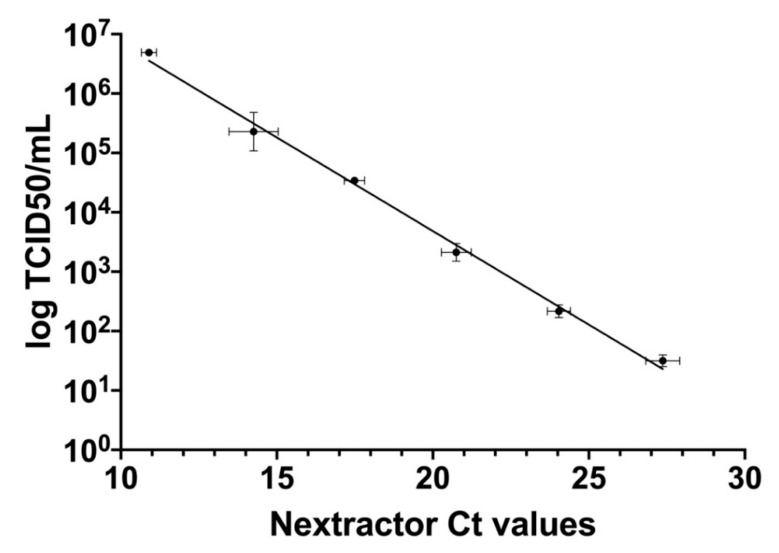
Correlation between TCID50/mL results (on a log10 scale) from titration on Vero E6 cells with the Reed and Muench method and Ct values derived from qRT-PCR with preliminary RNA extraction (Nextractor). Ct values are presented as the mean of six replicates for each dilution ± SE, whereas TCID50/mL values are presented as the mean of the two different methods (Reed and Muench and Karber) of two replicates for each dilution ± SE.

**Figure 6 viruses-13-01022-f006:**
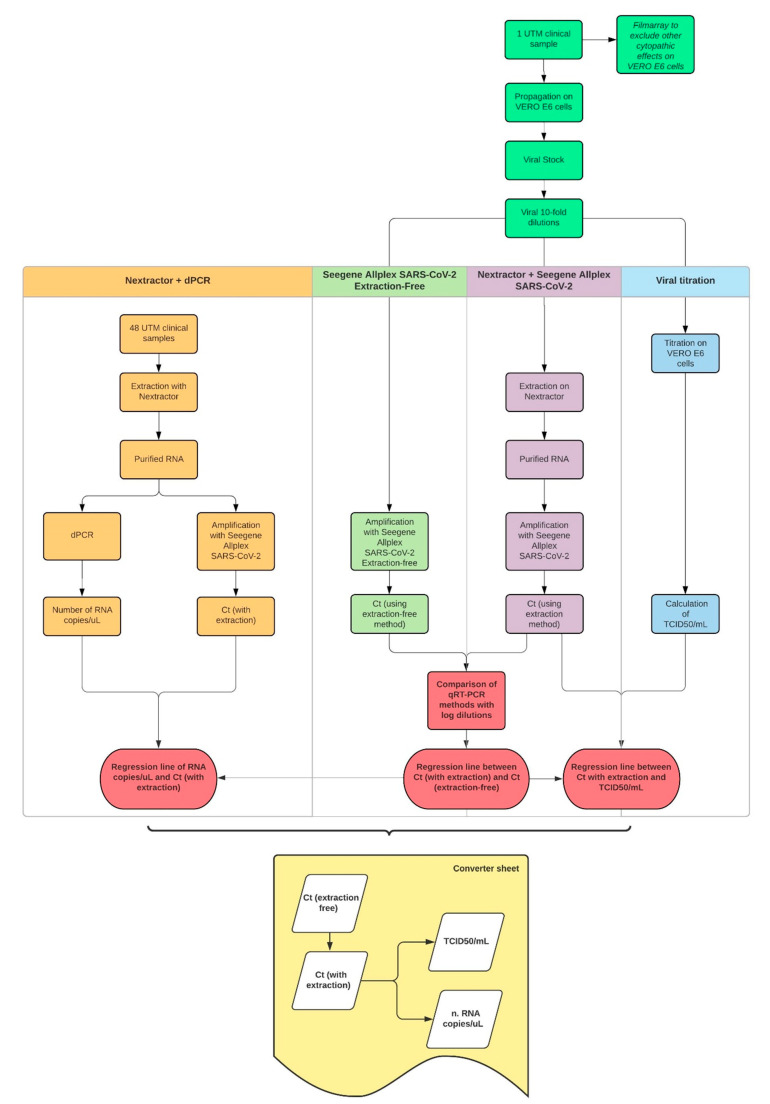
The workflow explaining the procedures and protocols used to correlate our results and obtain the converter sheet, the aim of the entire experiment.

## Data Availability

The data presented in this study are available on request from the corresponding author.

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
