# Peer review of "Correlating qRT-PCR, dPCR and Viral Titration for the Identification and Quantification of SARS-CoV-2: A New Approach for Infection Management"

_viruses, 2021, doi:10.3390/v13061022_

Round 1
Reviewer 1 Report
Revision manuscript: viruses-1197977
The manuscript is interesting and well-structured. It compares the results obtained using different analytical methods for detection and quantification of the novel Coronavirus (SARS-CoV-2). Two PCR methods were used (a qRT-PCR protocol (with or without extraction) and a digital PCR) and compared with a viral titer obtained in cell culture. The PCR methods provide information of the viral loads (presence and quantification of the RNA or part of it) and the cell culture add also the information on the infectiveness of the virus (as cytopathic effects on the cell culture). Thus, the methods provide different information on the specimens. The methods results are specified as TCID50/mL (cell culture), Ct values (qRT-PCR method) and number of RNA copies per µL (digital PCR). The present results showed a strong correlation among the latter mentioned method results. Additional information needs to be added on the samples (type and number) used in the study and results needs to be discussed more in detailed.
Introduction:
- Nice introduction on the three zoonotic Coronaviruses. It is complete and exhaustive.
- Information on the three analytical methods are given (two qRT-PCR protocols, a dPCR and cell cultures)
- Line 80: add abbreviation BSL-3, since the abbreviation is used afterwards in line 211
- Line 83: Please add specifications on the sentence, in particular on the types of variabilities connected with the use of a biological system (cell culture) and how they could affect the results
Material and methods:
- Line 122: The correlation between qRT-PCR and dPCR was obtained using residual clinical specimens. Please specify is these are human samples and the origin/type of sample (swabs, serum,..). In line 186 nasopharyngeal and oral swabs are mentioned. How many samples were used? If the RNA was present how this was preserved.
- Line 124: The correlation between molecular methods and endpoint titration was achieved using a viral strain isolated from a residual sample taken from a patient with laboratory-confirmed COVID-19. Please specify type of sample and on how many samples was performed.
- Line 135: what was the source of the viral stock?
- Line 176: “contemplates” seems to be not the correct verb in this sentence
- Line 185: dPCR was separately performed on 48 residual UTM from positive clinical samples 185 (nasopharyngeal and oral swabs). This was performed in addition to the other samples as with the qRT-PCR (viral stock silutions?).
- Line 186: were these samples previously quantified by qRT-PCR? please specify.
Results:
- Line 218: Ct values obtained processing each sample. Are these the residual clinical sample or the dilution of the viral stock?
- Line 261: Copies/µL (on a log10 scale) estimated by dPCR were graphed as a function of qRT- PCR Ct values obtained processing samples with preliminary extraction with Nextractor. Correlation using dPCR and the extraction-free protocol (Allplex SARS-CoV-2 Extraction-Free) was done?
- Line 267: qRT-PCR Ct values obtained processing samples with preliminary extraction with Nextractor and following amplification accordingly to Allplex SARS-CoV-2 Extraction- Free protocol were further correlated with TCID50/mL. same for the protocol without extraction?
- How was the correlation for the dPCR and the TCID50/mL?
Discussion:
- How you explain the constant off-set of the distribution among the qRT-PCT (with and without extraction)?
- How you explain the decreased correlation for Ct values above 27 the correlation decreases? (line 260-261)
- Line 279: Ct values from semi-quantitative methods can only be considered as a macroscopic approximation of the real viral burden. Please better explain this concept.
- Line 287: the latter, on the contrary, could lead to an overestimation of the titer by detecting the total amount of RNA molecules, including both genomic RNA and subgenomic mRNAs, rather abundant in clinical samples as well as in cell cultures. What control can be used for this in diagnostic settings?
- Line 295: These findings have been concretized in a converter sheet capable of returning an estimated value of RNA copies and TCID50/mL for a given Ct value obtained with Allplex SARS-CoV-2. However still both methods will be needed in diagnostic settings or not?
- Line 299:TCID50/mL thus determined can only be considered as a mere estimation due to the intrinsic variability of the method and the operator. Please relate this with the comment written for line 83
- In the title of the paper “a new approach for infection management” is mentioned. Is this the sole use of PCR in diagnostic setting without the use of cell culture. Please specify the correlation between the title and the last part of the discussion of this manuscript.

Author Response
Dear Reviewer,
thank you for your feedback, please find attached our responses to your observations.
Sincerely,
The authors

Reviewer 2 Report
Dear authors,
Your manuscript is concise, well written and aims to collaborate to the interpretation of the laboratory diagnostic of SARS-CoV-2 which is greatly appreciated. Few comments bellow:
Line 129 - It seems to me the authors were careful enough to test for other respiratory viruses the original sample used to isolate and propagate SARS-CoV-2 used in their assay. I imagine the authors did the same with the propagate, so I suggest adding here that information to the manuscript. If not I would recommend doing it and adding the result to the manuscript. Despite being very likely the propagate will confirm positive for SARS-CoV-2 and negative for the other respiratory viruses, testing the propagate as well would demonstrate an extra precaution.
Moreover, the virus used for dilutions was confirmed positive for SARS-CoV-2 by different molecular protocols, so I wonder if this isolate was ever sequenced for lineage identification. If so, please add the lineage to the text.
Line 185 - What does UTM stand for? I imagine it should be the acronym VTM that stands for viral transport media. If so, please correct here.
For the comparisons, the authors used a viral propagate in serial dilutions which is the most adequate thing to do. However, I would also like to see if an original clinical sample, not from propagate, submitted to a plaque assay would have the same results. It is very similar, but it would evaluate an original infectious virus in a clinical sample and not from a propagate that has already replicated in Vero cell culture. The chosen sample would be a sample with low ct more likely to have infectious virus and the sample would be titrated as the authors have done with the propagate. I'm aware the authors used 48 clinical samples to compare qRT-PCR and dPCR, but sounds these samples were not submitted to virus isolation and the infectivity status is unknown or unclear.
Also, reading this manuscript I got curious to know if the results would be the same using inactivated samples. That could be done using a clinical sample submitted a plaque assay as aforementioned, or even with the same propagate used in the analysis reported in this manuscript. Dilutions already confirmed with infectious virus would be submitted to an inactivation process and then tested by the same molecular methods to see if the results would be the same.
Finally, not all samples with low Cts will have infectious viruses, right? Because of that, just to make sure there is no misleading to the readers maybe a more conservative interpretation of the results would be indicated. It is true that according to the results infectious dilutions were related to molecular results, but it is not clear if inactivated or badly stored samples with low Cts would have the same correlation.
Author Response

(The authors gave the same response as above.)

Reviewer 3 Report
see the attached file

Author Response

(The authors gave the same response as above.)

Round 2
Reviewer 3 Report
Authors answered only in part to comments and suggestions. This manuscript needs to be re-written. At present it is at its infancy.